# Audio General Recognition of Partial Discharge and Mechanical Defects in Switchgear Using a Smartphone

Dongyun Dai [1,2], Quanchang Liao [1,2,*], Zhongqing Sang [1,2], Yimin You [1,2], Rui Qiao [1,2] and Huisheng Yuan [1,2]

[1] School of Electrical Engineering and Automation, Xiamen University of Technology, Xiamen 361024, China;
dongy82@163.com (D.D.); 18121079156@163.com (Z.S.); youyimin@126.com (Y.Y.);
2122031383@s.xmut.edu.cn (R.Q.); yuanhuisheng2023@163.com (H.Y.)

[2] Xiamen Key Laboratory of Frontier Electric Power Equipment and Intelligent Control, Xiamen 361024, China

* Correspondence: 2122031377@s.xmut.edu.cn; Tel.: +86-18250714950

**Abstract:** Mechanical defects and partial discharge (PD) defects can appear in the indoor switchgear of substations or distribution stations, making the switchgear a safety hazard. However, traditional acoustic methods detect and identify these two types of defects separately, ignoring the general recognition of audio signals. In addition, the process of using testing equipment is complex and costly, which is not conducive to timely testing and widespread application. To assist technicians in making a quick preliminary diagnosis of defect types for switchgear, improve the efficiency of the subsequent overhaul, and reduce the cost of detection, this paper proposes a general audio recognition method for identifying defects in switchgear using a smartphone. Using this method, we can analyze and identify audio and video files recorded with smartphones and synchronously distinguish background noise, mechanical vibration, and PD audio signals, which have good applicability within a certain range. When testing the feasibility of using smartphones to identify three types of audio signal, through characterizing 12 sets of live audio and video files provided by technicians, it was found that there were similarities and differences in these characteristics, such as the autocorrelation, density, and steepness of the waveforms in the time domain, and the band energy and harmonic components of the frequency spectrum, and new combinations of features were proposed as applicable. To compare the recognition performance for features in the time domain, frequency band energy, Mel-frequency cepstral coefficient (MFCC), and this method, feature vectors were input into a support vector machine (SVM) for a recognition test, and the recognition results showed that the the present method had the highest recognition accuracy. Finally, a set of mechanical defects and PD defects were set up for a switchgear, for practical verification, which proved that this method was general and effective.

**Keywords:** switchgear; PD; mechanical vibration; anomaly detection; audio signal processing; SVM; pattern recognition



## 1. Introduction

Switchgear play an important role in power systems as widely used electrical equipment for circuit isolation and protection against overloads and system faults [1]. During long-term operation, various defects in a switchgear inevitably occur. Studies have shown that the defects of a switchgear mainly include insulation defects, mechanical defects, and overheating defects [2]. Insulation defects are usually accompanied by PD, including corona discharge, suspension discharge, and internal discharge [3]. Mechanical defects include conductor contact loosening, shield loosening, and bolt loosening [4]. Either type of defect can lead to power system failures and cause serious economic losses. Therefore, the timely detection of possible defects and hidden dangers in the operation of a switchgear can ensure the reliability and safety of power system operation [5].

For the detection of a single defect type, PD can be detected via acoustic [3,6] and electrical methods [7,8], while mechanical defects are mainly detected via vibration [1,4]

and acoustic methods [9,10]. However, when an abnormality occurs in power equipment, selecting which defect detection method to use first is a problem. Although it is feasible to identify defect types on a case-by-case basis, the complexity of the process and the installation of specialized inspection equipment is not conducive to the initial identification of defects. To enhance detection efficiency, the literature [11,12] uses composite sensors for the simultaneous detection of PD and mechanical defects, but the preparations cost for this type of sensor are high, and there have been few applications in practical situations. All these methods have advantages and disadvantages for detecting switchgear defects, considering their complexity and cost of use, and acoustic detection has the advantages of inexpensive sensors, anti-electromagnetic interference, easy detection, and monitoring the operation of equipment online [13]. If PD and mechanical defects can be detected and identified using an acoustic method synchronously, this would be more economical, convenient, and applicable compared to other methods.

Currently, the use of acoustics to detect and identify mechanical defects or PDs in electrical equipment is being researched, and good progress has been made. In [9], fully integrated empirical mode decomposition was applied to the audio signals of induction motors, and mechanical defects were successfully detected after analyzing the edge frequency characteristics. In [10], acoustic pressure sensors were used to obtain audio intensity cloud maps of mechanical defects, and five eigenvalues extracted using a gray-level covariance matrix (GLCM) were used to realize the diagnosis of mechanical defects in a gas-insulated metal-enclosed switchgear (GIS). In [14], the accurate recognition of acoustic emission (AE) signals of electric tree branches was achieved using an artificial neural network (ANN) and SVM. In [15], MFCC features were extracted for AE signals, and an SVM classifier optimized using sequential minimum optimization (SMO) was used to accurately identify four types of PD defect in GIS.

In the above study, the acoustic detection and identification of mechanical defects used audio signals at low frequency, while PD defects used AE signals at high frequency; however, the two defects were detected and identified separately. In [3], a comprehensive identification method was proposed to distinguish the AE signals of PD and background noise, but background noise was not counted as a defect type. To enhance inspection efficiency and reduce costs, the method of using the recording function of smartphones to identify PD and mechanical defects of switchgear simultaneously seems feasible. However, few papers have used smartphones to record the audio of switchgear and identify defects. A related comparison is the use of smartphones for diagnosing mechanical defects in induction motors [16]. Although previous articles have proposed methods to characterize mechanical defects or PDs, it is important to investigate whether PDs and mechanical defects in switchgear can be identified directly and simultaneously from audio signals captured by smartphones. Therefore, this paper performed a study and provides a convenient method for defect detection and audio recognition. When a switchgear is defective and emits abnormal audible sound waves [17], this can assist technicians, even those who do not have a basic knowledge of acoustics, to use their smartphones to make a preliminary diagnosis of the type of defect in the switchgear and effectively identify mechanical vibration or PD for a subsequent evaluation with more specialized testing equipment.

To solve the appeal problem and to verify the feasibility of using the recording function of smart phones for the simultaneous detection and identification of mechanical defects and PDs in switchgear, the relevant work was carried out as follows: First, in view of the popularity of smartphones and the increasing power of audio functions, the data source was set to be the audio files captured by the microphone of smartphones. The 12 sets of audio files in this paper were provided by the technicians of substations or distribution stations, and according to the type of audio signal, they can be classified into three categories: namely, background noise, mechanical vibration, and PD. The three types of audio signals were analyzed in terms of time-domain features, such as waveform autocorrelation, denseness, and steepness, and frequency-domain features, such as band energy and harmonic components, and a new combination of features suitable for identifying these three types of audio

is proposed. Second, recognition tests were conducted using an SVM multiclassification support vector machine for time-domain features, frequency band energy features, and MFCC features from the literature [15], as well as the present method. The best features were obtained by comparing the accuracy of different features for recognizing mechanical vibration, PD, and background noise. Third, based on the above work, this paper proposes an audio generalized recognition method for PD and mechanical defects of a switchgear using smartphones.A set of mechanical defects and PD defects were set for the switchgear, to practically verify the generalization and effectiveness of the method.

This paper is organized as follows: Section 2 sequentially describes the acoustic theory of defects, the sources and types of audio file, the data preprocessing, and the methods of analysis and identification. Section 3 analyzes the characteristics of the audio data from the time and frequency domains and summarizes the features. The recognition accuracy of time domain, frequency band energy, MFCC features, and the present method on audio signals is compared using an SVM classifier. Section 4 gives the experimental validation of the method. Section 5 summarizes the research content and significance of this paper and proposes future work.

## 2. Materials and Methods
### 2.1. Acoustic Theory
2.1.1. Mechanical Vibration

In a normal state, the operating mechanism of switchgear and the mechanical condition of each electrical component is good. The current-carrying conductor generates mechanical vibration and emits sound waves under the interaction of electrodynamic forces, and the sound waves are recorded as normal vibration.

Suppose two current-carrying conductors in the switchgear are fed with sinusoidal AC currents $i_1, i_2$, assuming $i_1 = i_2$, where $i_1$ is:

$$i_1 = I_m \sin wt \tag{1}$$

In Equation (1), $I_m$ is the maximum value of current, $w$ is the angular frequency, and $t$ is the time.

Assuming that a single current-carrying conductor is subjected to an electromotive force $F_0$, the loop coefficient is $K_c$, and the cross-sectional coefficient is $K_h$, and the electromotive force between current-carrying conductors $F_0$ is

$$F_0 = \frac{\mu_0}{4\pi} K_c K_h i^2 \tag{2}$$

In Equation (2), $\mu_0$ is the vacuum magnetic permeability.
Substitute Equation (1) into Equation (2) to obtain

$$F_0 = \frac{\mu_0}{8\pi} K_c K_h I_m^2 (1 - \cos 2wt) \tag{3}$$

From Equation (3), when the frequency of the AC power supply is 50 Hz, the vibration frequency of the electrodynamic force between the current-carrying conductors is 100 Hz. The vibration frequency of both single-phase insulated and three-phase insulated GIS is 100 Hz [4]. Therefore, the fundamental frequency of the normal vibration of a switchgear is 100 Hz, without a harmonic component.

When there are mechanical defects, such as a loose component, poor contact, or deformation, the electrodynamic force will excite this switchgear with mechanical defects. Due to the non-linear phenomenon of the equipment response, the normal vibration that has changed is called abnormal vibration. Therefore, the fundamental frequency of abnormal vibration is 100 Hz and contains a harmonic component.

### 2.1.2. PD

The mechanism of PD is where the electric field strength in the local area is higher than the dielectric strength, and thus a discharge occurs, resulting in the deterioration of the dielectric. Over time, this will gradually erode the insulation medium and eventually lead to the failure of the insulation system [18]. The causes of discharge are related to the reduction in dielectric strength, such as dust invasion, moisture, and aging of the insulation; a second cause is related to the local electric field strength, such as over-voltage and a metal tip [5].

When PD occurs, air expands rapidly under the combined effect of pulsed electric field forces and thermal effects, which in turn causes the surrounding medium to vibrate violently. The discharge is usually in the form of a short period of continuous pulse generation, which causes a high number of air vibrations in a short period of time and triggers audible sound waves or even ultrasonic waves.For different types of discharge, the moment of discharge is different during the power frequency (50 Hz) cycle, which leads to different frequencies of acoustic pulse groups. The frequency of the acoustic pulse groups can be categorized into the following three cases:

- If both the positive and negative half-weeks are discharged, then the frequency of the acoustic pulse group is 100 Hz, such as the discharge along the surface and the internal air gap discharge;
- If only the positive or negative half-perimeter is discharged, the frequency of the acoustic pulse group is 50 Hz, such as metal tip discharge;
- In other cases, the acoustic pulse group may not have a specific frequency, such as free metal particle discharge.

Since the response of the gas is also nonlinear, the fundamental frequency of the audio signal of the discharge may be 50 Hz or 100 Hz with a harmonic component, or there may be no fundamental frequency and no significant harmonic component.

### 2.2. Data Selection

The data used in this paper were live audio and video files provided by technicians inside certain substations and distribution stations. During their indoor inspections or routine checks, they found abnormal audible sound waves emitted from the operating switchgear or the surrounding environment, and they subsequently used the audio or video recording function of their smartphones to obtain audio and video files. In addition, the technicians, after further dismantling and defect detection, provided actual results and some on-site photographs, as shown in Figure 1 below.

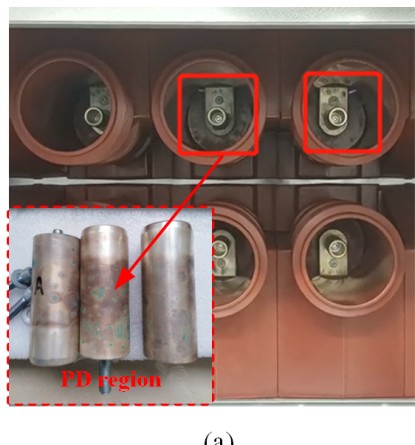　　　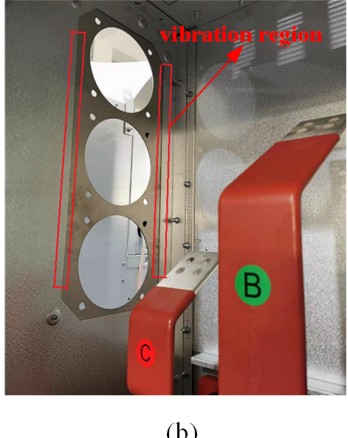

(a)　　　　　　　　　　　　　　　　　　　(b)

**Figure 1.** Actual defects in smartphone recording. (**a**) PD in a three-phase busbar contact box of switchgear; (**b**) Abnormal mechanical vibration in the side plate of a switchgear.

The audio and video files provided were all over 10 s in length, using smartphones that are widely available, and the built-in microphone sampled at 48 kHz or 44.1 kHz. According to the detection results, the files can be classified into three types of audio: background noise, mechanical vibration, and PD. Twelve typical audio files were selected and numbered one by one in capital letters, as shown in Table 1, with the different switchgear briefly indicated by numbers.

The four background noises come from different indoor environments, and the sound sources were human speech, metallic clang, birds chirping, and passing vehicle sounds. The mechanical vibration and PD came from the defects of the GIS, as well as other switchgears. The four mechanical vibrations contained normal vibration and abnormal vibration, and the abnormal vibrations came from actual mechanical defects, while the sound sources were current transformers (CTs), sheet metal parts, and side plates. Among the four PDs, two were from actual PD defects—the pressure relief flap and bus contact box discharges—and two were from simulated PD defects—single metal tip and two metal tip discharges.

**Table 1.** Sound source of audio files A1–C4.

| Audio Type | Audio File Number | Sound Source |
| --- | --- | --- |
| Background noise | A1 | Human speech |
| | A2 | Metallic clang |
| | A3 | Birds chirping |
| | A4 | Vehicle passing sounds |
| Mechanical vibration | B1 | Switchgear 1: Normal vibration |
| | B2 | GIS: CT abnormal vibration |
| | B3 | Switchgear 1: Abnormal vibration of sheet metal parts |
| | B4 | Switchgear 2: Abnormal vibration of side plate |
| PD | C1 | Switchgear 3: Discharge of the pressure relief flap |
| | C2 | Switchgear 4: Discharge of two metal tips |
| | C3 | Switchgear 5: Discharge of busbar contact box |
| | C4 | Switchgear 6: Discharge of single metal tip |

*2.3. Data Pre-Processing*

Due to the large amount of data in the audio file, to facilitate the analysis of the data and feature extraction, it was necessary to preprocess the data, and the steps included frame splitting, normalization, and adding window functions.

The first step was to split the frames. The audio file was intercepted with a duration of $1s$, which is called frame $x_i$, $i$ is the current number of frames, and the amount of data per frame $N$ is equal to the sampling frequency per second.

The second step was normalization. The size of the audio data varied depending on the acquisition process and the external environment. To avoid attributes in the larger value range dominating attributes in the smaller value range, the data were mapped to the $[-1, 1]$ interval. The normalized $x_i$ is

$$x_i(n) = \frac{x_i(n) - x_{mean}}{x_{max} - x_{min}}, 1 \leq n \leq N \tag{4}$$

$x_{max}$ is the maximum value in each frame, $x_{min}$ is the minimum value in each frame, and $x_{mean}$ is the average value in each frame.

The third step was to add window functions. To reduce the effect of spectral leakage of the Fourier transform, a Hamming window was added to smooth a frame of data, and the window function was

$$w(n) = 0.54 - 0.46 \times \cos(\frac{2\pi n}{N}) \tag{5}$$

Then, the audio data $y_i$ after adding the window were

$$y_i(n) = x_i(n) \times w(n) \tag{6}$$

*2.4. Data Analysis Methods*

2.4.1. Correlation Analysis

To compare the correlation between two signals or vectors, the cosine similarity formula was used for calculation. Suppose there are two vectors of dimension $K$ with $g$ and $l$, respectively. Their correlation coefficients are $S_{gl}$ with values in the range $(-1, 1)$, 1 being positive correlation and $-1$ being negative correlation.

$$S_{gl} = \frac{\sum_{k=1}^{K} g(k)l(k)}{\sqrt{\sum_{k=1}^{K} g(k)^2} \times \sqrt{\sum_{k=1}^{K} l(k)^2}}, 1 \leq k \leq K \tag{7}$$

2.4.2. Frequency Domain Analysis

The frequency domain amplitude spectrum was obtained using fast Fourier transform. Since the sampling frequency of the microphone was 48 kHz, the upper frequency limit of the spectrum was 24 kHz in theory, but in the actual measurements, the upper frequency limit in the spectrum may be three- to four-times lower than the sampling frequency, and the upper frequency limit of the signal in this paper was 16 kHz–17 kHz. The Fourier transform equation is

$$Y_i(f) = \sum_{n=1}^{N} y_i(n)e^{-i\frac{2\pi f}{N}n}, f = 1, 2, \cdots, N \tag{8}$$

The frequency interval 0–16 kHz of the amplitude spectrum was divided into 16 frequency bands in the order from small to large, and the length of the band was 1 kHz and was recorded as $E_j$. The energy of the band $E_j$ is defined as the sum of the frequency amplitude of the band and the formula is

$$E_j = \sum_{f=1000(j-1)}^{1000j} Y_i(f), 1 \leq j \leq 16 \tag{9}$$

The band vector $T$ was constructed and normalized for 16 bands of energy, which was used to represent the energy share of different bands, where $E$ is the sum of energy in the frequency band 0–16 kHz, $E = \sum_{1}^{16} E_j$, and the expression is

$$T = [\frac{E_1}{E}, \frac{E_2}{E}, \cdots, \frac{E_{16}}{E}] \tag{10}$$

*2.5. Support Vector Machine*

An SVM finds an optimal hyperplane in the hyperspace, where the set of input parameters is located to achieve the classification of different sets of input parameters. Each set of input parameters can be described by using a single-dimensional or multidimensional input vector and is suitable for solving complex pattern recognition problems with small sample data, high feature dimensionality, and nonlinearity [19,20].

Figure 2 has two types of linearly separable data, represented by circles and triangles, described by vectors $d = d_1, d_2$. The two types of data can be accurately distinguished by solving for the optimal hyperplane $wd + b = 0$, where $w$ is the normal vector of the hyperplane and $b$ is the distance between the hyperplane and the origin. For a linearly indistinguishable set of input parameters, the input vector is mapped to a high-dimensional feature space using a kernel function, and the hyperplane is obtained in that space, thus transforming it into a linearly divisible problem [1].

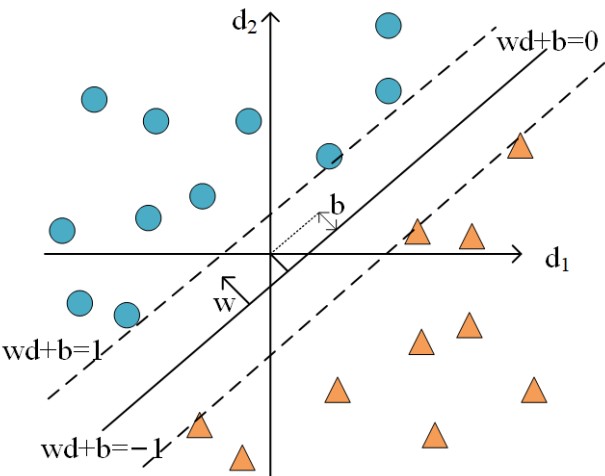

**Figure 2.** SVM classification principle.

## 3. Audio Characterization and Recognition Tests

### 3.1. Data Analysis

The microphones and filters integrated into smartphones have a restricted performance, contributing to a restricted frequency range and audio signal reproduction. In addition, there are different types of mechanical defects and PDs in the switchgear, as well as various types of background noise. To be able to accurately identify the three types of audio data recorded by smartphones, this section analyzes the similarities and differences in the direction of time domain waveforms, frequency band energy, and harmonic components.

#### 3.1.1. Continuous Time Stability

To evaluate the stability of time domain and frequency domain features of a single audio file in continuous time, audio data with a duration of 10 s were selected for correlation analysis. The correlation coefficients of time domain amplitude and frequency domain amplitude of 10 frames were calculated using Equation (7).

Let us define $S_{ij}$ as the time domain amplitude correlation coefficient of frame $y_i$ and frame $y_j$, and define $F_{ij}$ as the frequency domain amplitude correlation coefficient of frame $y_i$ and frame $y_j$, where $1 \leq i \leq j \leq 10$. The maximum and minimum values of the correlation coefficients were taken for analysis, and the results are shown in Table 2.

The correlation coefficients of the 10 frames' time domain amplitude were lower than 0.313 for any two frames of the background noise and PD, and the adjacent frames showed no correlation, while for the mechanical vibration, the correlation coefficients of the adjacent frames reached 0.956, but the non-adjacent frames showed no correlation. This shows that the waveforms of background noise and PD were not similar, and the amplitude changes were irregular, while the waveforms and amplitude of mechanical vibration were more similar in continuous time.

The correlation coefficients of the 10 frames' frequency domain amplitude were higher than 0.94 for normal vibration, 0.732 for abnormal vibration, 0.462–0.884 for PD, and 0.102–0.857 for background noise. This shows that the frequency domain amplitude distribution of mechanical vibration and PD were similar in continuous time, while the frequency domain amplitude distribution of background noise was less similar.

In terms of feature stability in continuous time, the time domain features of mechanical vibration were more stable, the frequency domain features of mechanical vibration and PD were more stable, whereas the background noise was less stable in both time and frequency domain features.

**Table 2.** Similarity calculation of 10 frames of data generated a symmetric matrix of $10 \times 10$ similarities, from which the minimum value and the maximum value other than 1 were selected.

| Audio Type | Audio File Number | $min(S_{ij})$ | $max(S_{ij})$ | $min(F_{ij})$ | $max(F_{ij})$ |
|---|---|---|---|---|---|
| Background noise | A1 | $-0.063$ | 0.108 | 0.102 | 0.692 |
| | A2 | $-0.065$ | 0.036 | 0.347 | 0.784 |
| | A3 | $-0.396$ | 0.313 | 0.521 | 0.838 |
| | A4 | $-0.097$ | 0.094 | 0.635 | 0.857 |
| Mechanical vibration | B1 | $-0.84$ | 0.835 | 0.94 | 0.978 |
| | B2 | $-0.134$ | 0.913 | 0.863 | 0.997 |
| | B3 | $-0.212$ | 0.345 | 0.732 | 0.895 |
| | B4 | $-0.06$ | 0.956 | 0.806 | 0.994 |
| PD | C1 | $-0.113$ | 0.173 | 0.731 | 0.847 |
| | C2 | $-0.103$ | 0.155 | 0.462 | 0.884 |
| | C3 | $-0.023$ | 0.024 | 0.75 | 0.781 |
| | C4 | $-0.023$ | 0.028 | 0.764 | 0.801 |

### 3.1.2. Time Domain Waveform Shape

To analyze the similarities and differences of the same type but with different sound sources and different audio types, one frame was randomly selected from the 10 frames of the audio file as a typical frame of the audio file. The first 100 ms of the time domain waveforms of the typical frames of the three types of audio are shown in Figures 3–5.

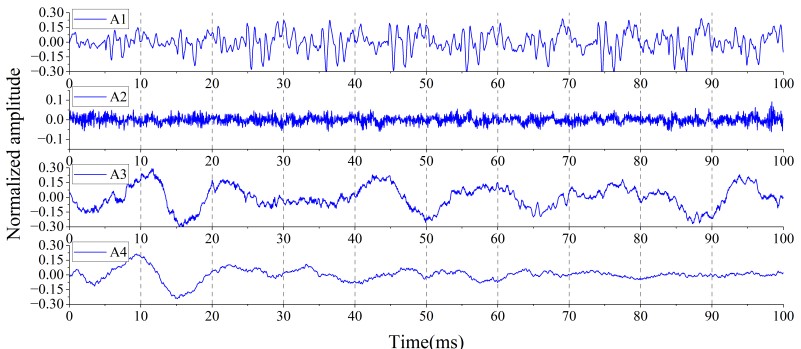

**Figure 3.** Time domain waveform of background noise.

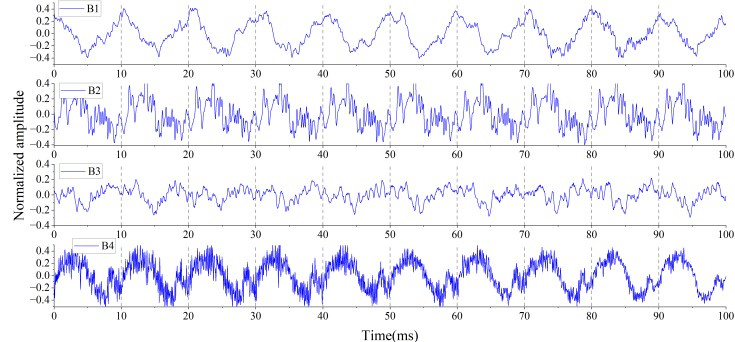

**Figure 4.** Time domain waveform of mechanical vibration.

In Figure 3, the background noise, A1–A4 have no similar waveforms, and the waveforms are irregular. In Figure 4, mechanical vibration, B1–B4 have 10 similar waveforms and the time interval is about 10 ms, the waveforms are sparse and gentle, and the shapes are like sine waveforms. In Figure 5, the waveforms of C1–C4 are dense and steep, C2 and C4 have acoustic pulse groups with pulse intervals of about 20 ms and 10 ms, respectively, C1 and C3 have no acoustic pulse groups and correlation, and the waveforms vary irregularly.

The analysis shows that the time domain waveform of background noise is usually irregular and does not have autocorrelation. The mechanical vibration generated by the operating switchgear has an regular waveform pattern, stable amplitude, and autocorrelation. Although acoustic pulse clusters appear in the PD about every 10 ms or 20 ms, the occurrence and duration of the acoustic pulse clusters and the amplitude changes are not the same, resulting in a low autocorrelation of the time-domain waveforms.

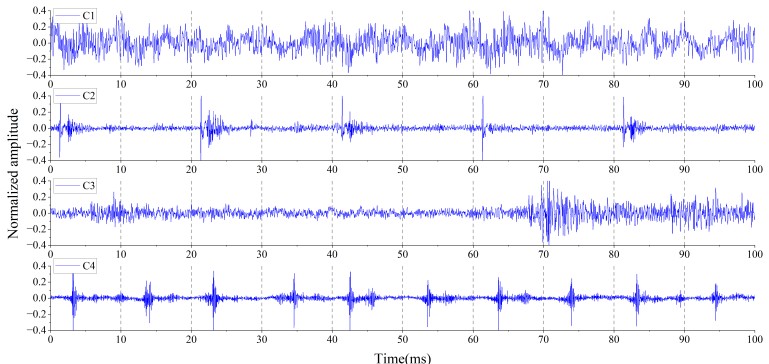

**Figure 5.** Time domain waveform of PD.

### 3.1.3. Frequency Band Energy Share and Harmonic Component

To analyze the frequency energy share of the three types of audio in different frequency bands, the frequency band vectors T of A1–C4 were calculated and compared, as seen in Figure 6. A1, A3, A4, and B1–B3 have similar energy shares, with the largest energy share at 0–1 kHz, rapidly decreasing to less than 10% at 0–5 kHz, and stabilizing after 5 kHz. A2, B4, and C1–C4 have similar energy shares, with the energy share in each band not exceeding 15%. Unlike the previous decreasing trend, the energy share of these audios after 5 kHz shows an increasing and then decreasing trend. In addition, the energy share of C2 and C4 shows another increasing trend after 10 kHz. The spectra of A2, B4 and C1–C4 were further analyzed, and it was found that A2 has high frequencies only part of the time within a frame-length window, while B4 and C1–C4 have continuous high frequencies throughout the window.

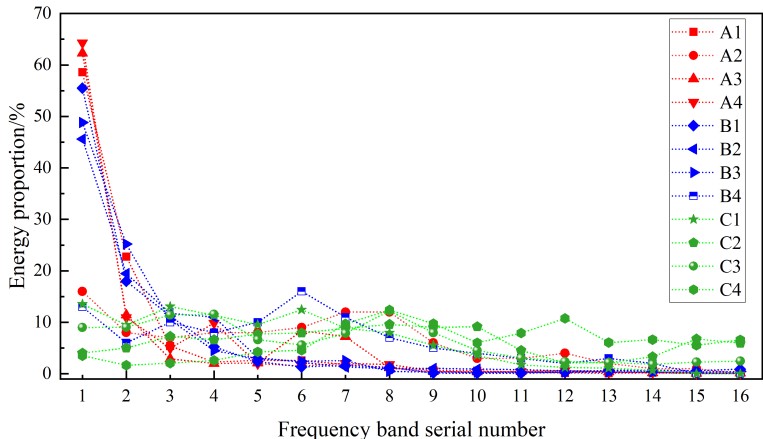

**Figure 6.** Energy share of different frequency bands for a typical frame.

This analysis showed that most of background noise and mechanical vibration were distributed in the low frequency band of 0–5 kHz, and the energy share of different frequency bands decreased with the increase in frequency, while the energy share of PD in 0–16 kHz was balanced, and the energy was more concentrated in the high frequency band of 5 kHz–16 kHz . However, some background noise and mechanical vibrations were also distributed in the frequency band above 5 kHz, with a similar energy share as PD,

with the difference being that the high frequency duration of the background noise was short. Although PD has a higher frequency distribution and can be used to distinguish between background noise and mechanical vibrations, the limited microphone performance of smartphones restricts the acquisition of PD signals above the 16 kHz band. Therefore, the accuracy of identifying the three types of audio captured with smartphones from the band energy alone may not be high.

In Figure 7, the maximum amplitude values of background noise A1–A4 appear at 203 Hz, 52 Hz, 60 Hz, and 78 Hz, respectively, with neither a fundamental frequency nor significant harmonic component. In Figure 8, the maximum amplitude values of mechanical vibration B1–B4 all appear at the fundamental frequency of 100 Hz, but the harmonic components are different. B1 has no harmonic component, and the harmonic components of B2 are mainly distributed at 300 Hz, 600 Hz, and 800 Hz, while the B3 is distributed at 200 Hz, 600 Hz, and B4 is distributed at 200 Hz. In addition, B4 still has 100 Hz harmonic component in the frequency band of 5 kHz–8 kHz . In Figure 9, there is no harmonic in C3, whereas harmonic components appear in C1, C2, and C4. The fundamental frequency of C4 is 100 Hz, but C1 and C2 are 50 Hz. In addition, the frequency of the maximum magnitude does not coincide with the fundamental frequency. The maximum magnitude of C1, C2, and C4 are 100 Hz, 800 Hz, and 54 Hz, respectively.

This analysis shows that there was no fundamental frequency and harmonic component in the frequency domain spectrum of the background noise, and if there was a fundamental frequency, it was not 50 Hz or 100 Hz. The four sets of mechanical vibrations in the paper came from different vibration types from the switchgear. Although the fundamental frequency was 100 Hz and the fundamental values were all equal to the maximum magnitude, the harmonic components were different. The harmonic component of abnormal vibration was large, but there was almost no harmonic component to the normal vibration. Therefore, the difference in harmonic components can be used as a feature for smartphones to recognize different mechanical vibrations in the switchgear. The frequency domain spectrum of PD has three cases, including a 100 Hz fundamental frequency, 50 Hz fundamental frequency, and no fundamental frequency, which correspond to a time domain waveform with a 10 ms or 20 ms period of acoustic pulse group, or no periodic acoustic pulse.This comparison reveals that PDs with 100 Hz fundamental frequency and mechanical vibration are not distinguishable in terms of harmonic components. Neither the PD without fundamental frequency nor the background noise had harmonic components and they were indistinguishable, and only the PD with 50 Hz fundamental frequency was distinguishable.

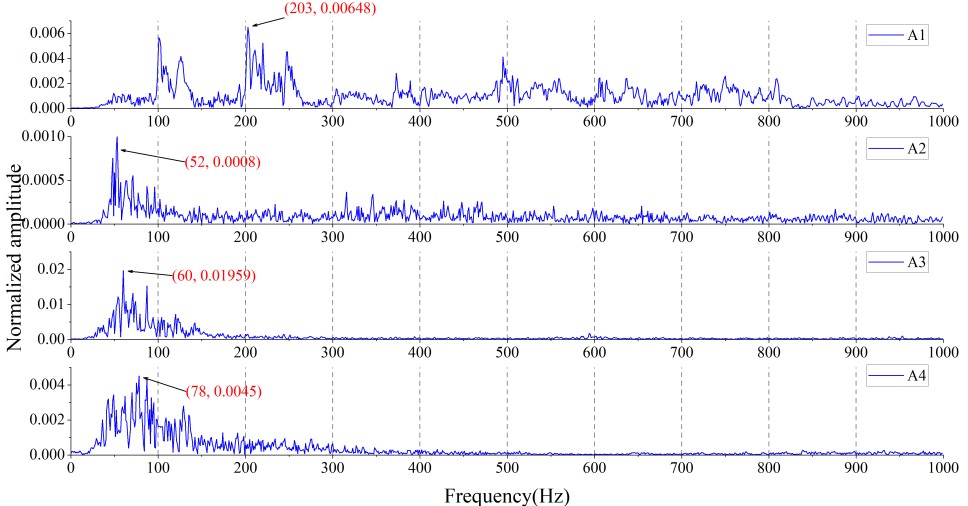

**Figure 7.** Background noise frequency domain spectrum.

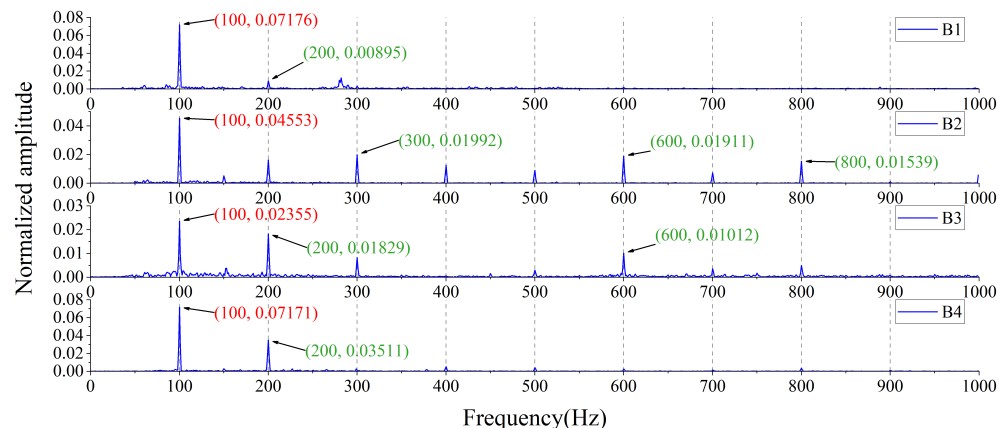

**Figure 8.** Mechanical vibration frequency domain spectrum. The red colour represents the maximum amplitude and the green colour represents the main harmonic components.

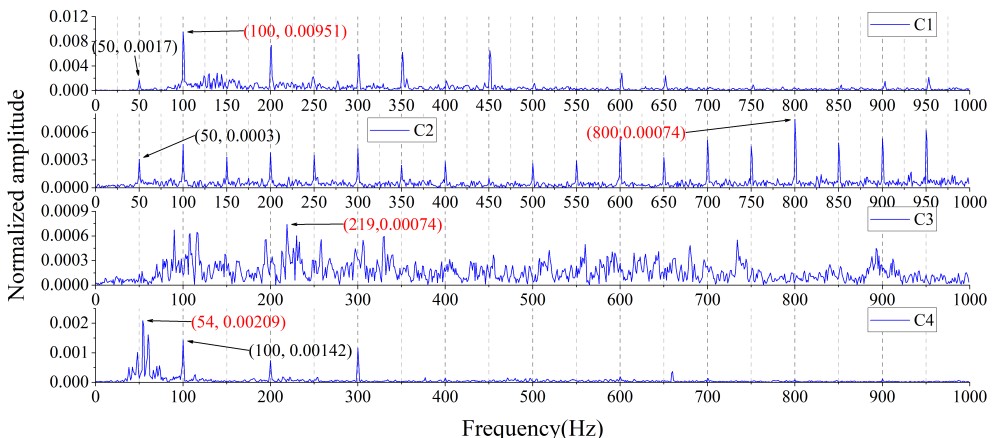

**Figure 9.** PD frequency domain spectrum.

### 3.2. Acoustic Feature Differences and Feature Selection

After analyzing the time domain waveforms and frequency domain spectra, the three types of audio captured by smartphones had the following similarities and differences. Table 3 shows the differences in the three types of audio for the time domain characteristics, and Table 4 shows the differences in the three types of audio for the frequency domain characteristics.

**Table 3.** Differences in Time Domain Characteristics.

| Audio Type | Correlation | Waveform Density | Waveform Steepness |
|---|---|---|---|
| Background noise | No | Low/High | Smoother |
| Normal vibration | Yes | Low | Flat and gentle |
| Abnormal vibration | Yes | Low/High | Smoother |
| PD | No | High | Steep |

**Table 4.** Differences in frequency domain characteristics.

| Audio Type | Frequency Band Energy Distribution | Fundamental Frequency | Frequency Harmonics | Maximum Amplitude at Fundamental Frequency |
|---|---|---|---|---|
| Background noise | 0–5 kHz/5 kHz–16 kHz | No/non-50 Hz and 100 Hz | No | No |
| Normal vibration | 0–5 kHz | 100 Hz | No | Yes |
| Abnormal vibration | 0–5 kHz/5 kHz–16 kHz | 100 Hz | Yes | Yes |
| PD | 5 kHz–16 kHz | No/50 Hz/100 Hz | No/Yes | No/Yes |

As shown in Table 3, five features $d_1$–$d_5$ in the time domain were selected, as also shown in Table 5. $d_1$ calculates the number of times the signal passes the zero value in each frame, $d_2$ and $d_3$ describe the direction of waveform skew and the sharpness of waveform kurtosis, $d_4$ describes the extremity of the waveform, and $d_5$ measures the correlation of the waveform within each frame. In Table 5, $y(n)$ is the time domain data of each frame, $\mu$ represents the mean value of $y(n)$, and the formula $\mu = E[y(n)]$, $\sigma$ is the mean squared deviation, $\sigma = \sqrt{E(y(n)^2) - \mu^2}$. $d_5$ first divides $y(n)$ into 10 equal segments and uses Equation (7) to calculate the correlation coefficient among the 10 segments.

**Table 5.** Selected time domain features and formulas.

| Feature Number | Feature Name | Formula |
|---|---|---|
| 1 | Short-time over-zero rate | $d_1 = \sum_{n=2}^{N} \|sgn[y(n)] - sgn[y(n-1)]\|, sgn[y] = \begin{cases} 1, y \geq 0 \\ 0, y < 0 \end{cases}$ |
| 2 | Skewness | $d_2 = \frac{E[(y-\mu)^3]}{\sigma^4}$ |
| 3 | Kurtosis | $d_3 = \frac{E[(y-\mu)^4]}{\sigma^4}$ |
| 4 | Peak factor | $d_4 = \frac{max(\|y(n)\|)}{\sqrt{\frac{1}{N}\sum_{n=1}^{N} y(n)^2}}$ |
| 5 | Correlation coefficient | $d_5 = \frac{y_i(n) \times y_j(n)}{\|y_i(n)\| \times \|y_j(n)\|}, 1 \leq i \leq j \leq 10$ |

According to Table 4, five frequency domain features $v_1$–$v_5$ were selected, as shown in Table 6. Among them, $v_1$ determines whether the average frequency energy is concentrated in 5 kHz–16 kHz, $v_2$ and $v_3$ determine whether the frequency energy is concentrated in the 50 Hz or 100 Hz component in 0–1 kHz, and $v_4$ and $v_5$ evaluate the 100 Hz harmonic component in 0–1 kHz. $Y(f)$ in Table 6 is the amplitude of the audio signal at frequency $f$. The prerequisite for calculating $v_4$ is to determine whether the maximum amplitude in 0–1 kHz is at 100 Hz, and if it is, then it is calculated, otherwise it is recorded as 0. $max(Y(100f))$ represents the maximum amplitude of the 100 Hz harmonic in 0–1 kHz.

**Table 6.** Selected frequency domain features and formula.

| Feature Number | Feature Name | Formula |
|---|---|---|
| 1 | Average High-frequency share | $v_1 = \sum_{f=1}^{10}(\sum_{f=5000}^{16,000} Y(f) / \sum_{0}^{16,000} Y(f)) \times \frac{1}{10} \times 100\%$ |
| 2 | 100 Hz component share | $v_2 = \sum_{f=1}^{10} Y(100f) / \sum_{0}^{1000} Y(f) \times 100\%$ |
| 3 | 50 Hz even/odd harmonic sum | $v_3 = \frac{\sum_{f=2}^{20} Y(50f)}{\sum_{f=1}^{19} Y(50f)}, f = \begin{cases} 2, 4, \cdots, 20 \\ 1, 3, \cdots, 19 \end{cases}$ |
| 4 | Harmonic maximum value/fundamental frequency value | $v_4 = \frac{max(Y(100f))}{Y(100)} \times 100\%, f = 2, 3, \cdots, 10$ |
| 5 | Fundamental frequency value/harmonic sum | $v_5 = Y(100) / \sum_{f=2}^{10} Y(100f), f = 2, 3, \cdots, 10$ |

For the recognition of different audio types, the correct selection of features and computation is a very important issue. Too few and too many features do not signify a high recognition rate, but in fact only features that characterize significant differences should be selected. Harmonic components are effective for identifying different types of mechanical vibration, but PDs also have similar harmonic components. At the same time, there is a

similar percentage of energy in the frequency band for the three types of audio. Therefore, it was necessary to combine the above features, and the combination of features proposed in this paper is notated as $TF = (d_5, v_1, v_2, v_3, v_4, v_5)$.

### 3.3. SVM Classifier Training and Recognition

#### 3.3.1. Identification of a Single Audio Type

To verify the performance of the present feature vector TF in identifying a single audio type, the first step was to extract the TF features of 207 frames of data from A1–C4. The normal vibration needed to be identified separately in some cases, so three and four types of labels were added to create four sets of feature libraries, corresponding to SVM classifiers TF1 and TF2. In addition, the time domain features, $d = (d_1, d_2, d_3, d_4, d_5)$, corresponding to the classifiers CD1 and CD2, and the frequency band vector, T, corresponding to the classifiers CT1 and CT2, and the MFCC features proposed in the literature [15] , corresponding to the classifiers CM1 and CM2, were jointly compared. Next, the classifier was configured with the penalty parameter set to 1, the kernel function was a Gaussian kernel, and the decision function type was one-to-one. Then, the data of the feature library were divided into two parts in a ratio of 7:3, and the training set was used as the input vector of the classifier for training, and finally the training and test sets were cross-validated, and the average score was calculated [19].

The recognition rates of the classifiers are shown in Table 7. TF1 and TF2 had the highest total recognition accuracies of 99.6% and 98.6%, respectively, while the recognition accuracy for CD2 was only 88%, due to its inability to recognize normal vibrations. CM1 and CM2 had better recognition rates for mechanical vibration and PD, but lower recognition accuracy for background noise, at 89.4% and 92.4%, respectively. CT1 and CT2 had the lowest total recognition accuracy, with some of the PDs being misclassified as vibrations, and failing to recognize normal vibrations. Therefore, compared to time domain features, MFCC features, and frequency band energy features, the proposed TF features had the best recognition performance and could recognize background noise, normal vibration, abnormal vibration, and PD in SVM.

**Table 7.** SVM classifier recognition accuracy.

| Features | Classifier | Recognition Accuracy (%) | | | | |
| | | Background Noise / 66 Frames | Normal Vibration/ 19 Frames | Abnormal Vibration /58 Frames | PD/ 64 Frames | Total/ 207 Frames |
|---|---|---|---|---|---|---|
| Time domain | CD1 | 100 | | 96.1 | 96.8 | 97.8 |
| | CD2 | 100 | 0 | 94.8 | 96.8 | 88 |
| TF | TF1 | 100 | | 98.7 | 100 | 99.6 |
| | TF2 | 100 | 100 | 93.1 | 100 | 98.6 |
| MFCC | CM1 | 89.4 | | 97.7 | 100 | 95.7 |
| | CM2 | 92.4 | 100 | 98.2 | 100 | 96.2 |
| frequency band energy | CT1 | 83.3 | | 96.1 | 96.8 | 92.2 |
| | CT2 | 86.3 | 0 | 94.8 | 96.8 | 83.7 |

#### 3.3.2. Defective Audio Identification with Different Noise Components

To study the recognition of mechanical vibration and PD with different background noise components, first, 10 frames from A1–C4 were selected and normalized; then, the background noise with different components was superimposed with mechanical vibration and PD, to generate 160 frames of mechanical vibration and 160 frames of PD with background noise. Finally, CD1 and TF1 were used to identify the defective audio. The recognition results are shown in Figure 10.

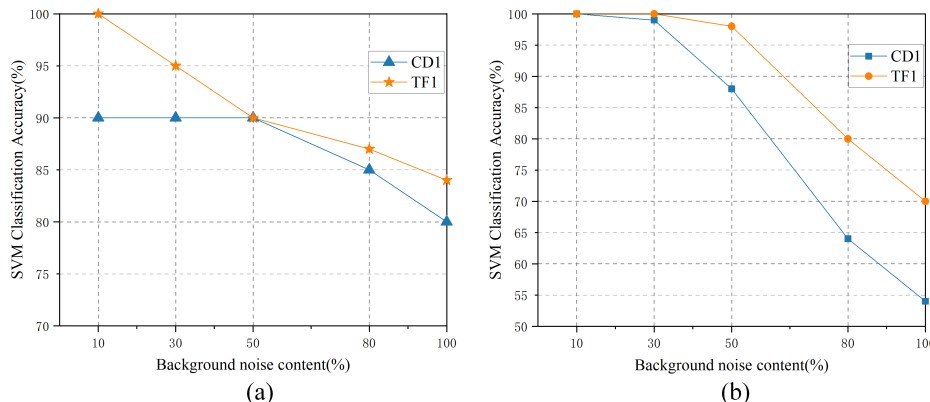

**Figure 10.** Accuracy of defective audio recognition under different noise components. (**a**) Mechanical vibration; (**b**) PD.

In Figure 10a, when the background noise component of mechanical vibration was at or below 50%, the recognition rates of CD1 and TF1 remained above 90%, and with the increase in the background noise component, the recognition rates of CD1 and TF1 finally decreased to 80% and 84%. In Figure 10b, TF1 maintained an 80% recognition rate when the background noise component of PD reached 80%, while the recognition rate of CD1 decreased rapidly to 54% as the background noise component increased.

This analysis shows that, when the switchgear had mechanical defects or PD defects and the indoor background noise component was large, the proposed TF features combined with the SVM classifier could still identify the mechanical vibration and PD better; compared with the selected time domain features, the proposed TF features also showed a better classification performance.

## 4. Experimentation and Verification

To verify the effectiveness of the present method for recognizing PD in an indoor environment, experiments were conducted in the insulation withstand voltage experimental platform shown in Figure 11a. This experimental platform consisted of an AC voltage withstand test system, a test transformer, and a capacitive voltage divider, which could generate a voltage of 0–100 kV for simulating the occurrence of PD. A PD defect was set inside the switchgear, and then the switchgear was connected to the experimental circuit. At the same time, a smartphone was placed on the side 40 cm away from the switchgear for recording audio files, and the sampling rate of the smartphone was 48 kHz. The background noise was recorded for 15 s before powering up the experimental platform, and then the voltage level of the experimental platform was gradually increased. When the voltage level reached 7.2 kV, the PD produced audible sound waves for 30 s, then we reduced the voltage and ended the recording. The total duration of the recording was 60 s and was recorded as Audio 1.

In addition, to verify that the present method could also recognize the audio signals of mechanical vibrations, a mechanical defect was set up in the switchgear cabinet shown in Figure 11b. When the switchgear was operated under power, the mechanical defect could be clearly noticed as generating vibrations accompanied by audible sound waves. The handheld smartphone first recorded 15 s of indoor background noise and then 45 s of mechanical vibration signals 40 cm closer to the switchgear, and the total duration of the recording was also 60 s, recorded as Audio 2.

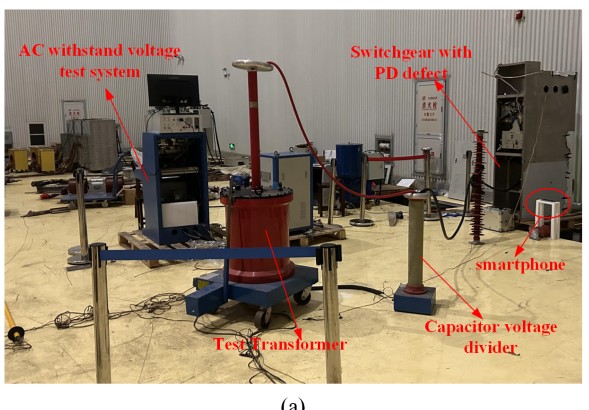 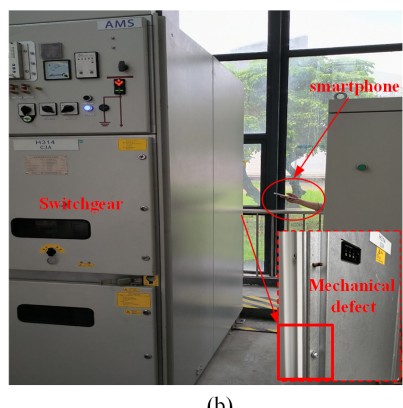

(a)                                                             (b)

**Figure 11.** Experimental platform for identifying switchgear defects using smartphones. (**a**) PD defect and experimental circuits; (**b**) Experimental circuits for mechanical defects, which were loose screws.

The recognition was performed on two sets of audio files on a PC, and the recognition process started with data preprocessing; then, the frequency domain feature $v_1-v_5$ were extracted for each frame and finally input into the SVM classifier TF1 for recognition, and the results are shown in Figure 12. For audio 1, the recognition value from the 21st to the 51st second was 2, which corresponds to PD, and the rest of the recognition values were 0, which corresponds to background noise. This recognition result was consistent with the experimentally set discharge duration of 30 s. For audio 2, the recognition value from the 15th to the 60th second was 1, which corresponds to mechanical vibration, and the rest of the recognition values were 0, which corresponds to background noise. This identification result was in line with the 45 s vibration duration of the experimental setup. The above experiments demonstrated that the present method could accurately identify background noise, mechanical vibration, and PD from audio signals.

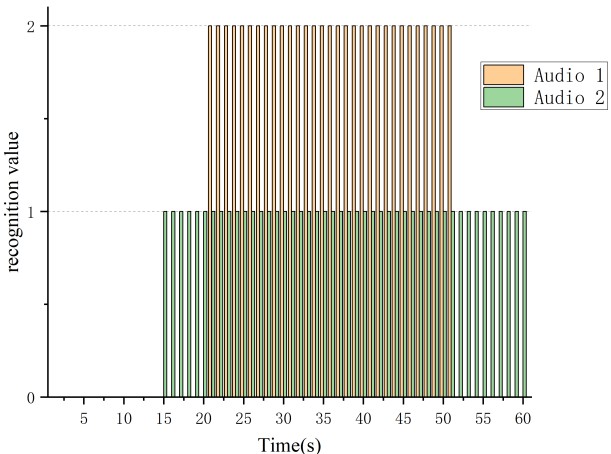

**Figure 12.** Recognition results for audio 1 and 2. The recognition result for background noise was 0, the recognition value for mechanical vibration was 1, and the recognition result for PD was 2.

## 5. Conclusions

This paper proposed a method for identifying defects in a switchgear using audio from smartphones. First, audio files including background noise, mechanical vibration, and PD were selected and preprocessed. Then, the similarities and differences of each type of audio signal, in terms of time domain waveforms, frequency band energies, and harmonic components, were studied. Then, the time domain, frequency band energy features, MFCC features, and the present method were extracted and input into a SVM for recognition and to compare the accuracy. Finally, the method was experimentally validated, and the following conclusions were obtained:

- Currently, widely available smartphones record audio in a frequency range of about 0–16 kHz, which makes the distinction between the three types of audio signals in terms of frequency band energy potentially small. Mechanical vibrations and PDs of switchgear may both have harmonic components, but there are differences in the fundamental frequency and harmonic distributions. In addition, mechanical vibration has a high autocorrelation in the time domain waveform, while background noise and PD have a low autocorrelation. Accurate identification of the three types of audio requires a combination of these features;
- In the recognition test for the time domain, frequency band energy, MFCC features, and the present features combined with SVM and comparison of the recognition rate, it was found that the present features had the highest accuracy in recognizing background noise, mechanical vibration, and PD, which provides a new idea for the screening of audio features for mechanical defects and PD;
- The popularity of smartphones makes audio files easily accessible. In the experiment, by applying this method to recognize the audio recorded with a smartphone, the result proved that this method could well recognize three types of audio signal. This method can help technicians to rapidly diagnose the defects of a switchgear, and it has good versatility and applicability.

The number of selected audio files in this paper was limited, and the audio signals of special working conditions, such as switchgear breaking and closing operations, were not considered. Therefore, more types of background noise, mechanical vibration, and PD need to be collected, and the feature and recognition algorithm should be optimized to improve the recognition accuracy for various audio signals of a switchgear.

**Author Contributions:** Conceptualization, D.D.; methodology, D.D. and Q.L.; software, Q.L. and H.Y.; validation, Q.L. and Z.S.; formal analysis, Y.Y.; data curation, R.Q. and Y.Y.; writing—original draft preparation, Q.L.; writing—review and editing, R.Q.; funding acquisition, D.D. All authors have read and agreed to the published version of the manuscript.

**Funding:** This work was supported in part by the Xiamen Key Laboratory of Frontier Electric Power Equipment and Intelligent Control, Xiamen, 361024, China.

**Institutional Review Board Statement:** Not applicable.

**Informed Consent Statement:** Not applicable.

**Data Availability Statement:** Data available on request due to restrictions eg privacy or ethical. The data presented in this study are available on request from the corresponding author. These data are not disclosed to the public due to the commercial secrecy of the enterprise's production activities.

**Conflicts of Interest:** The authors declare no conflict of interest.

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
