# Peer review of "Audio General Recognition of Partial Discharge and Mechanical Defects in Switchgear Using a Smartphone"

_applsci, doi:10.3390/app131810153_

Round 1

Reviewer 1 Report

The reliability of the proposed method for detection of two types of malfunctions (abnormal vibration, partial discharges) in switchgear has been verified on a small data set (12 audio recordings), thus the suitability of the methodology for practical deployment is not demonstrated. This is admitted by the authors themselves in the conclusion, so further experimental validation and optimization of the proposed method will be needed. Also, the sensitivity to noise in the recording was verified on synthetic data, not on real recordings, and there is therefore room for further development here as well. Nevertheless, I consider the results of this work promising. Formally, I would recommend modifying the graph in Figure 5, it is not very readable.

Reviewer 2 Report

There is not any imaging of the experimental setup.

There is not any comparison with previous studies.

Reviewer 3 Report

The proposed work was intended to investigate “Research and Recognition of Acoustic Characteristics of Partial Discharge and Mechanical Defects in Switchgear”. However, minor revision is required considering the following points mentioned below:  

1.     Revision of abstract is suggested to include proper introduction to problem.  

2.     Literature review techniques have to be strengthened by including the issues in the current system and how the author proposes to overcome the same.

3.     The results should be compared and discussed with other papers published previously.

4.     Corrections of some grammatical errors and typos are needed.

Corrections of some grammatical errors and typos are needed.

Reviewer 4 Report

1) I suggest that the Authors describe in more detail the contribution of their work to the field of PD and mechanical fault recognition in switchgear. Both the acoustic method and the classification technique based on SVM have been known and effectively used for many years. The two presented research problems (lines 56-63) have not been supported by any references that would indicate that these are problems that are difficult to solve and require additional investigations.

2) Line 116-119: "When the sinusoidal AC voltage is near or at the peak of the positive and negative half-cycle, the insulation system is subjected to the maximum electric field strength, and the probability of discharge is the greatest."

This statement is a bit imprecise because the moment of ignition of the discharge depends on its type. In the case of creeping/surface discharges or discharges in internal gas inclusions, their ignition takes place before the peak of the positive and negative half-cycle, i.e. in the first and third half of the AC voltage sine wave.  

3) The labels of the audio files in Table 2 (A1, A2,..., C4) and in Graphs 2-4 (a1, a2, ..., c4) need to be unified. The same problem is in the text of Section 3.1.2.

4) How were the time and frequency domain features selected? According to numerous works on PD recognition, the use of such frequency domain features as Spectral Centroid, Weighted Peak Frequency, Partial Power, or Spectral Skewness gives very good classification results. Have the authors tested these features?

5) The article lacks information about the measurement system used and the procedure for recording audio signals generated by the tested defects.

Reviewer 5 Report

1-    The paper’s title should be revised. It seems the research can be removed in the paper’s title.

2-    The innovations of the paper should be highlighted in the paper’s title. SVM can be used in the paper’s title?

3-    The English language of the text should be revised.

4-    The sentences should be shortened. For instance, the first sentence of the abstract is so long. It can be divided into separate sentences.

5-    It is not good to express that switchgears develop the PD and acoustic emissions. It is better that PD and AE might appear in switchgears.

6-    PD sound? It seems the paper should be checked carefully to use better expressions for these concepts. Or defective switchgear is not meaningful.

7-    The present sentences should be used in the abstract. “was” has been used in the abstract, which is not OK.

8-    The literature review should be improved by citing and reviewing recent relevant references.

9-    Some recent papers that have focused on the practical cases are as follows:

A.     R. Zadeh and L. C. Heredia, "Partial Discharge Detection in High-Voltage Gas Insulated Switchgear Using Fiber Optic Based Acoustic Sensors," 2023 INSUCON - 14th International Electrical Insulation Conference (INSUCON), Birmingham, United Kingdom, 2023, pp. 216-220.

B.     A. R. Zadeh et al., "PD detection in inverter-fed machines using acoustic emission sensors.A preliminary investigation," 2022 IEEE Electrical Insulation Conference (EIC), Knoxville, TN, USA, 2022, pp. 280-283, doi: 10.1109/EIC51169.2022.9833214.

C.     Q. Lin, F. Lyu, S. Yu, H. Xiao and X. Li, "Optimized Denoising Method for Weak Acoustic Emission Signal in Partial Discharge Detection," in IEEE Transactions on Dielectrics and Electrical Insulation, vol. 29, no. 4, pp. 1409-1416, Aug. 2022, doi: 10.1109/TDEI.2022.3183662.

D.     L. Kirkcaldy, G. Lees, R. Rogers and P. Lewin, "Time Synchronized Distributed Acoustic Sensing of Partial Discharge at the Oil-Pressboard Interface," in IEEE Transactions on Dielectrics and Electrical Insulation, vol. 29, no. 6, pp. 2348-2353, Dec. 2022, doi: 10.1109/TDEI.2022.3203913.

E.     F. Witos, "Application of Acoustic Emission Method for Partial Discharge Research in Selected Elements and Devices of Electric Power Systems," 2018 Joint Conference - Acoustics, Ustka, Poland, 2018, pp. 1-6, doi: 10.1109/ACOUSTICS.2018.8502315.

10-                   Data selection is not clear. How this data has been collected. It should be clarified and explained in detail.

1-                   Normalizing the data is useful? Why? It makes sense?

1-                   Some quantitative advantages should be addressed in the abstract.

1-                   The methodology should be explained in more detail.

1-                   What is the advantage of this work compared to the available ones?

Extensive editing of English language required

Round 2

Reviewer 2 Report

It is clear that the PD and mechanical defects have different frequency domain contents and this fact does not need any classifier for distinguishing between PD and mechanical defects. Investigating each defect has been studied in different previous studies. This paper has not any new content.

Reviewer 4 Report

The Authors took into account all suggestions and comments. The manuscript has been improved and is fit for publication.

Reviewer 5 Report

The authors have tried to respond to review comments.

Minor editing of English language required
